# Robust Hybrid Learning With Expert Augmentation

**Antoine Wehenkel**                                          *awehenkel@apple.com*
*Apple*

**Jens Behrmann**                                          *j_behrmann@apple.com*
*Apple*

**Hsiang Hsu**                                          *hsianghsu@g.harvard.edu*
*Harvard*

**Guillermo Sapiro**                                          *gsapiro@apple.com*
*Apple*

**Gilles Louppe**                                          *g.louppe@uliege.be*
*University of Liège*

**Jörn-Henrik Jacobsen**                                          *jhjacobsen@apple.com*
*Apple*

**Reviewed on OpenReview:** *https://openreview.net/forum?id=oe4dl4MCGY*

## Abstract

Hybrid modelling reduces the misspecification of expert models by combining them with machine learning (ML) components learned from data. Similarly to many ML algorithms, hybrid model performance guarantees are limited to the training distribution. Leveraging the insight that the expert model is usually valid even outside the training domain, we overcome this limitation by introducing a hybrid data augmentation strategy termed *expert augmentation*. Based on a probabilistic formalization of hybrid modelling, we demonstrate that expert augmentation, which can be incorporated into existing hybrid systems, improves generalization. We empirically validate the expert augmentation on three controlled experiments modelling dynamical systems with ordinary and partial differential equations. Finally, we assess the potential real-world applicability of expert augmentation on a dataset of a real double pendulum.

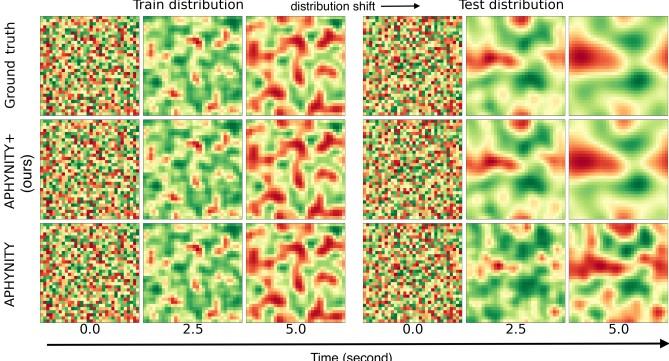

Figure 1: APHYNITY, an existing hybrid modelling strategy, is unable to predict accurately the dynamic of a 2D diffusion reaction for a shifted test distribution, although it predicts well configurations that follow the training distribution. APHYNITY+, the same model fine-tuned with our expert augmentation, generalizes to shifted distributions as expected from the validity of the underlying physics.

## 1    Introduction

Generalizing to unseen data is crucial to make a model applicable in the real world. When training and test data are independently and identically distributed (IID), we assess the model generalization on a held-out subset of the training data. Unfortunately, the training and test scenarios do not entirely overlap in practice. This observation has motivated many recent research efforts to focus on the robustness of ML models (Gulrajani & Lopez-Paz, 2020; Geirhos et al., 2020; Koh et al., 2021). Common strategies can be broadly grouped into two categories. The first class of methods aims to align properties of the model (e.g., invariance, equivariance or monotonicity) with expertise in the problem of interest (Cubuk et al., 2019; Mahmood et al., 2021; Keriven & Peyré, 2019; Silver et al., 2017). The second category is data-focused (Sagawa et al., 2019; Arjovsky et al., 2019; Krueger et al., 2021; Creager et al., 2021), and leverages variations present in the training data, e.g., some methods minimize the worst-case sub-group performance, to achieve robustness.

The data-oriented methods, which include Group-DRO (Sagawa et al., 2019) and Invariant Risk Minimization (Arjovsky et al., 2019, IRM), present the advantage to specify the invariances implicitly via domains or environments. However, these methods rely on variations in the training data, which may be insufficient when the problem is too complex, or the variations of interest are absent from the training set. On the other hand, methods based on domain-specific expertise do not suffer from such limitations. Embedding expertise into a model can be done via architectural inductive biases (LeCun et al., 1995; Xu et al., 2019), data augmentation (Cubuk et al., 2019), or a learning objective that enforces established symmetries of the problem (Cranmer et al., 2020). For example, simple data augmentation techniques combined with convolutions lead to excellent performance on natural image problems (Cubuk et al., 2019). Another natural approach to embedding expertise in ML models, and the one studied in this paper, is called hybrid learning, or sometimes gray-box modelling (Willard et al., 2022). This framework combines an expert model (e.g., from first principle physics) with a learned component that improves the expert model so that the combination better fits real-world data. In hybrid learning, the expert model plays a central role and is supposed to provide a simple and well-grounded (parametric) description of the process considered.

In recent works (Yin et al., 2021; Takeishi & Kalousis, 2021; Qian et al., 2021; Mehta et al., 2021; Lei & Mirams, 2021; Reichstein et al., 2019), hybrid learning demonstrated success in complementing partial physical models and improving the inference of the corresponding parameters. We observe that current hybrid learning algorithms are sub-optimal in the amortized inference setting – when we aim to build hybrid models that are valid for various test configurations. Contrary to the common belief that hybrid learning achieves better generalization than black box ML models, we argue and demonstrate that hybrid learning algorithms do not yet meet their promise regarding robustness in amortized settings. Although hybrid learning achieves strong performance on IID test distributions by exploiting the inductive bias of the expert models, their performance collapses when the test domain is not included in the training domain. This is unsatisfactory as the expert model is typically well-defined far outside the training distribution.

We introduce *expert augmentations* for training augmented hybrid models (AHMs), a procedure that extends the range of validity of hybrid models and improves generalization, as pictured by Figure 1. Our contribution is to first formalise the hybrid learning problem as: 1) Learning a probabilistic model partially defined by the expert model; 2) Performing inference over this probabilistic hybrid model. In this context, we show that hybrid learning is vulnerable to distribution shifts for which the expert model is well defined (see Figure 1, bottom row). Motivated by our analysis, we propose to fine-tune the hybrid model on an expert-augmented dataset that includes distribution shifts (see results of augmentation in Figure 1, middle row). These expert augmentations only rely on the hybrid model itself, leveraging that the expert model is also well-defined outside of the training distribution. Our experiments on various controlled problems demonstrate that AHMs improve the generalization capabilities of state-of-the-art hybrid learning algorithms on synthetic and real-world data in the amortize setting.

## 2    Hybrid learning

We formalize hybrid learning with the probabilistic model depicted in Figure 2, and later rely on this formalism to show the benefits of the proposed proposed expert augmentation. In this Bayesian network,

capital letters denote random variables (e.g., $Y$) and, we use calligraphic letters for the domain of the corresponding realization (e.g., $y \in \mathcal{Y}$). In our formalism, the expert model is a conditional density $p(y_e|x, z_e)$ that describes the distribution of the *expert* response $Y_e$ to an input $x$ together with a parametric description of the system $z_e$, denoting expert parameters. We augment the expert model with the *interaction model* which is a conditional distribution $p(y|x, y_e, z_a)$ that describes the distribution of the observation $Y$ given the input $x$, the expert model response $y_e$, and a parametric description of the interaction model $z_a$.

Our goal is to create a robust predictive model $p(y|x, (x_o, y_o))$ of the random variable $Y$, given the input $x$ together with independent observations $(x_o, y_o)$ of the same system, where the subscript $o$ denotes an observed quantity. As a concrete example, we consider predicting the evolution of a damped pendulum (described in Section 4.1) given its initial angle and speed ($x = [\theta, \dot{\theta}]$) and a sequence of observations of the same pendulum. The expert model we assume is able to describe a frictionless pendulum whose dynamic is only characterized by one parameter $z_e := \omega_0$, denoting its fundamental frequency. The expert model is misspecified; it does not model the friction with a second parameter $z_a := \alpha$, the damping factor. In this problem, $(x_o, y_o)$ and $(x, y)$ are IID realizations of the same pendulum which corresponds, in general terms, to samples from $p(x, y|z_a, z_e)$ for some fixed but unknown values of $z_a$ and $z_e$. The expert variables $z_e$ (e.g., $\omega_0$) together

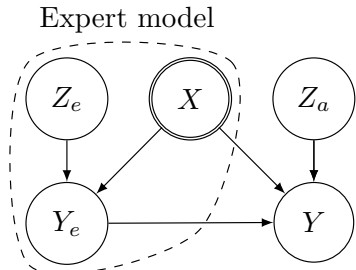

Expert model

Figure 2: A hybrid probabilistic model which describes the relationship between a given input $X$ and the output $Y$ for a configuration of the system as defined by the latent variables $Z_e$ and $Z_a$. The prescribed expert model defines the conditional density $p(y_e|z_e, x)$, where $Y_e$ is an approximation of $Y$. Hybrid learning aims at learning the conditional distribution $p(y|z_a, y_e, x)$.

with $z_a$ (e.g., $\alpha$) should accurately describe the system that produces $Y$ (e.g., the evolution of the pendulum's angle and speed along time) from $X$ (e.g., the initial pendulum's state). In our setting we assume that we are given a pair $(x_o, y_o)$ (e.g., past observations) from which we can accurately infer the state of the system $(z_a, z_e)$ as described by the interaction and expert models, and then predict the distribution of $Y$ for a given input $x$ (e.g., forecasting future observations) to the same system. Provided all probability distributions in Figure 2 are known, the Bayes optimal hybrid predictor $p_B$ is

$$p_B(y|x, (x_o, y_o)) = \mathbb{E}_{p(z_a, z_e|(x_o, y_o))} \left[ p(y|x, z_a, z_e) \right], \tag{1}$$

as shown in Appendix E.

In the amortized setting, we aim to learn a model of both the predictive model $p(y|x, z_a, z_e)$ and of the posterior over the parameters $p(z_a, z_e|(x_o, y_o))$. We will see that existing hybrid learning algorithms neglect the importance of building a robust encoder $p(z_a, z_e|(x_o, y_o))$ to make predictions in out-of-distribution (OOD) settings. In particular, we note that the interaction between $z_e$ and $y$ is essentially defined by the expert model. Thus it should be possible, and preferable, to learn a predictive model of $Y$ whose performance guarantees are as independent as possible from the training distribution of the expert variables $z_e$. Indeed, the validity of the expert model often go beyond the seen training examples. However, we demonstrate below that existing hybrid learning algorithms' performance collapses when the distribution of $z_e$ shifts. The expert augmentation introduced in this paper improves the robustness of hybrid models to such shifts.

## 2.1 Hybrid generative modelling

As common in the hybrid learning literature, we consider expert models that are deterministic although our discussion clearly extends to stochastic expert models. The expert model describes the system as a function $f_e : \mathcal{X} \times \mathcal{Z}_e \rightarrow \mathcal{Y}_e$ that computes the response $y_e$ to an input $x$, parameterized by expert variables $z_e$. The goal of hybrid modelling is to augment the expert model with a component learned from data as depicted in Figure 2. Formally, given a dataset $\mathcal{D} = \{(x^{(i)}, y^{(i)})\}_{i=1}^N$ of $N$ IID samples, we aim to learn the interaction model $p_\theta(y|x, y_e, z_a)$ that fits the data well but is close to the expert model. Defining closeness is hard and is an application-dependent modelling choice beyond the scope of this paper. However, common metrics include

L2-distance between expert and hybrid outputs or Kullback-Leibler (KL) divergence between the marginal distributions of $Y$ and $Y_e$.

Learning a model that is close to the expert model and fits the training data well is a hard problem. However, the APHYNITY algorithm (Yin et al., 2021) and the Hybrid-VAE (Takeishi & Kalousis, 2021, HVAE) are two recent approaches that offer promising solutions to this problem. We now briefly describe how these methods approximate the Bayes optimal predictor of Equation (1). Our augmentation strategy is compatible (and effective) with both approaches.

**APHYNITY.** Yin et al. (2021) consider hybrid learning for augmenting expert models defined as ordinary differential equation (ODE). They consider an additive hybrid model that should allow a perfect fitting of the data. In probabilistic terms, this assumption is equivalent to assuming the conditional distribution $p_\theta(y|x, y_e, z_a)$ is a Dirac distribution. Formally, they solve the following optimization problem

$$\min_{z_e, F_a} ||F_a|| \quad \text{s.t.} \quad \forall (x, y) \in \mathcal{D}, \forall t, \frac{dy_t}{dt} = F_e(y_t, z_e) + F_a(y_t)$$
$$\text{with} \quad y_0 := x, \tag{2}$$

where $|| \cdot ||$ is a norm operator on the function space, $F_a : \mathcal{Y}_t \to \mathcal{Y}_t$ is a learned function, $F_e : \mathcal{Y}_t \times \mathcal{Z}_e \to \mathcal{Y}_t$ defines the expert model and $\mathcal{D}$ is a dataset of initial states $x := y_0$ and sequences $y \in \mathcal{Y} := (\mathcal{Y}_t)^k$, where $k$ is the number of observed timesteps. APHYNITY solves this problem with Lagrangian optimization and Neural ODEs (Chen et al., 2018) to compute derivatives. In the context of ODEs, the random variable $X$ is the initial state of the system at $t_0$ and $Y$ is the observed sequence of $k$ states between $t_0$ and $t_1$.

This formulation only considers learning a missing dynamic for one realization of the system described by Figure 2, for a single $z_e$ (and $z_a$). However, we are interested in learning a hybrid model that works for the full set of systems described by Figure 2. As suggested in Yin et al. (2021), we use an encoder network $g_\psi(\cdot, \cdot) : \mathcal{X} \times \mathcal{Y} \to \mathcal{Z}_a \times \mathcal{Z}_e$ that corresponds to a Dirac distribution located at $g_\psi$ as the approximate posterior $q_\psi(z_a, z_e|x, y)$. The interaction model is a product of Dirac distributions whose locations correspond to the solution of the ODE

$$\frac{dy_t}{dt} = F_e(y_t, z_e) + F_a(y_t, z_a; \theta), \quad y_0 := x. \tag{3}$$

Hence the corresponding approximate Bayes predictor replaces the parameters $(z_a, z_e)$ in Equation (3) with the prediction of $g_\psi$ and predicts a product of Dirac distributions.

**Hybrid-VAE (HVAE).** In contrast to APHYNITY, the hybrid-VAE proposed by Takeishi & Kalousis (2021) is not limited to additive interactions between the expert model and the ML model, nor to ODEs. Instead, their goal is to learn the generative model described by Figure 2. They achieve this with a variational auto-encoder (VAE) where the decoder specifically follows Figure 2. Similarly to the amortized APHYNITY model, the encoder $g_\psi(x, y)$ predicts a posterior distribution over $z_a$ and $z_e$, and the model is trained with the classical Evidence Lower Bound on the likelihood (ELBO). Takeishi & Kalousis (2021) observe that relying only on an architectural inductive bias and maximum likelihood training is not enough to ground the generative model to the expert equations. They propose to add three regularizers $R_{PPC}, R_{DA,1},$ and $R_{DA,2}$ that encourage the generative model to rely on the expert model. The final objective is

$$\max_{\theta, \psi} \mathbb{E}_{\mathcal{D}} \left[ \text{ELBO}((x, y); \psi, \theta) \right] + \alpha R_{PPC} + \beta R_{DA,1} + \gamma R_{DA,2}. \tag{4}$$

The first regularizer, $R_{PPC}$, encourages the marginal distribution of samples generated by the complete model to be close to the marginal distribution that would be only generated by the physical model. The two other regularizers specifically require the encoder network for $z_e$ to be made of two sub-networks. The first network filters the observations to keep only what can be generated by the expert model alone; it predicts $z_a$ and remove its effect from $y$ to predict a filtered version of it. The second network maps the filtered observations to the posterior distribution over $z_e$. $R_{DA,1}$ penalizes the objective if the observations generated by the expert model are not close to the filtered observations. Finally, $R_{DA,2}$ relies on data augmentation

with the expert model to enforce that the second sub-network correctly identifies the expert variables $z_e$ if the observations were correctly filtered by the first encoder. We refer the reader to Appendix B and Takeishi & Kalousis (2021) for more details on HVAE. For HVAE, the approximate predictor takes the form described by Equation (1) where $p(z_a, z_e|(x_o, y_o))$ is approximated by the encoder $q_\psi(z_a, z_e|x, y)$ and $p(y|x, z_a, z_e)$ by the learned hybrid generative model.

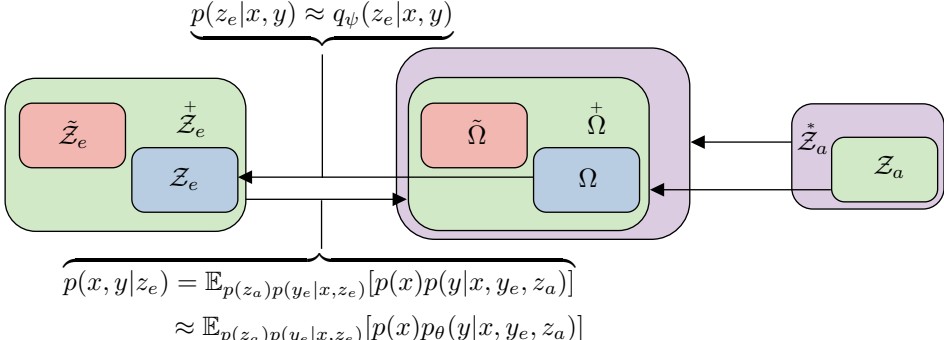

$$p(x, y|z_e) = \mathbb{E}_{p(z_a)p(y_e|x, z_e)}[p(x)p(y|x, y_e, z_a)]$$
$$\approx \mathbb{E}_{p(z_a)p(y_e|x, z_e)}[p(x)p_\theta(y|x, y_e, z_a)]$$

Figure 3: Visualization of the distribution shifts considered in this work. The train support $\Omega$ of $(x, y)$ results from $(z_a, z_e) \in \mathcal{Z}_a \times \mathcal{Z}_e$. The test supports (in red) are denoted with a tilde symbols as $\tilde{\mathcal{Z}}_e$ for $z_e$ and $\tilde{\Omega}$ for $(x, y)$. The augmented support $\overset{+}{\Omega}$ (in green) includes both train and test scenarios and corresponds to $(z_a, z_e) \in \mathcal{Z}_a \times \overset{+}{\mathcal{Z}}_e$. The outer violet domain that includes $\overset{+}{\Omega}$ depicts one of our experiment in which the domain of $z_a$ is also shifted. Hybrid modelling algorithms alone may learn a mapping $p_\theta : \overset{+}{\mathcal{Z}}_e \to \overset{+}{\Omega}$ but augmentation is necessary to learn the inverse mapping $q_\psi : \overset{+}{\Omega} \to \overset{+}{\mathcal{Z}}_e$.

## 3 Robust hybrid learning

We now formalize our definition of out of distribution (OOD) and robustness. In general, a test scenario is OOD if the joint test distribution $\tilde{p}(x, y)$ is different from the training distribution $p(x, y)$, that is $d(\tilde{p}, p) > 0$ for any properly defined divergence or distance $d$ between distributions. In the following, we reduce our discussion to a sub-class of distribution shifts for which the marginal train and test distributions over $z_e$ may be different, $d(p(z_e), \tilde{p}(z_e)) > 0$, but the marginals of $z_a$ and $x$ are constant. As a consequence, the joint distribution of $(x, y)$ pairs is also shifted. Formally, the training and test distributions are respectively defined as

$$p(x, y) := \mathbb{E}_{p(z_e)p(z_a)p(y_e|x, z_e)}[p(x)p(y|x, y_e, z_a)],$$
$$\tilde{p}(x, y) := \mathbb{E}_{\tilde{p}(z_e)p(z_a)p(y_e|x, z_e)}[p(x)p(y|x, y_e, z_a)].$$

In this context, we demonstrate, theoretically and empirically, that classical hybrid models fail. To address this failure, we introduce *augmented hybrid models* and show that, under some assumptions, they achieve optimal performance on both the train and test distributions.

Our goal is to learn a predictive model

$$p_{\theta, \psi}(y|x, (x_o, y_o)) = \mathbb{E}_{\underset{p(y_e|x, z_e)}{q_\psi(z_a, z_e|x_o, y_o)}}[p_\theta(y|x, y_e, z_a)]$$

that is *exact* on both the train and test domains when they follow the aforementioned training and testing distribution shifts. We say that a learned predictive model $\hat{p}(a|b)$ is $\mathcal{E}$-*exact*, or *exact* on the sample space $\mathcal{E}$, if $\hat{p}(a|b) = p(a|b) \quad \forall (a, b) \in \mathcal{E}$. Here we qualify a predictive model as *robust* to a test scenario if its *exactness* on the training domain is sufficient to ensure exactness on the test domain.

We now define an augmented distribution $\overset{+}{p}(z_e)$ over the expert variables whose support $\overset{+}{\mathcal{Z}}_e$ includes the joint support $\mathcal{Z}_e \cup \tilde{\mathcal{Z}}_e$ between the train and test distribution of the physical parameters. As depicted in Figure 3, we denote the corresponding support over the observation space $\mathcal{X} \times \mathcal{Y}$ as $\overset{+}{\Omega}$ for the augmented distribution, $\Omega$ for the training distribution, and $\tilde{\Omega}$ for the test distribution. In this context, and with **A1**, we

may demonstrate that even under perfect learning, classical hybrid learning algorithms do not produce an $\tilde{\Omega}$-*exact* predictor while our augmentation strategy does.

**Assumption 1 (A1):** *Hybrid modelling learns an interaction model $p_\theta(y|y_e, x, z_a)$ that is $\overset{+}{\Omega}$-exact.*

Although strong, **A1** is consistent with the recent literature on hybrid modelling, which assumes that the expert model $p(y_e|x, z_e)$ is an accurate description of the system, thereby the interaction model $p_\theta(y|y_e, x, z_a)$ should not be overly complex. As an example, we consider an additive interaction model in our experiments for which extrapolation to unseen $y_e$ holds if the additive assumption is correct. That said, we still notice that the exactness of the interaction model $p_\theta$ on the augmented support $\overset{+}{\Omega}$ is insufficient to prove that the predictive model $p_{\theta,\psi}$ is $\overset{+}{\Omega}$-*exact*. Indeed, the encoder $q_\psi$ is only trained on the training data and cannot rely on a strong inductive bias in contrast to $p_\theta$. Thus, even if the encoder is exact on the training distribution, the corresponding predictive model is not $\overset{+}{\Omega}$-*exact*. While the decoder's performance are not limited to the training scenarios thanks to the broader validity of the expert model, the encoder does not generalize to unseen settings as it is purely data-driven.

### 3.1 Expert augmentation

We propose a data augmentation strategy to improve the robustness of hybrid models to unseen test scenarios. Once trained, the hybrid model is composed of an encoder $q_\psi$ and an interaction model $p_\theta$ that are respectively $\Omega$- and $\overset{+}{\Omega}$-*exact*. We may create a new training distribution with a support over $\overset{+}{\Omega}$ by sampling physical parameters $z_e$ from a distribution that covers $\overset{+}{\mathcal{Z}}_e$. Then, we train the encoder $q_\psi$ on the augmented domain $\overset{+}{\Omega}$, under perfect training the predictive model $p_{\theta,\psi}(y|x, (x_o, y_o))$ is $\overset{+}{\Omega}$-*exact*, hence exact on both train and test domains. The expert augmentation is formally described in Algorithm 1, for further details see Appendix A.

Our learning strategy is grounded in existing hybrid modelling algorithms, and here, we focus on APHYNITY and HVAE. We first train an encoder $q_\psi$ and a decoder $p_\theta$ with a hybrid learning algorithm. Together with experts we then decide on a realistic distribution $\overset{+}{p}(z_e)$ and create a new dataset $\overset{+}{\mathcal{D}} := \{(z_e, x_i, y_i)\}_{i=1}^{\overset{+}{N}}$ by sampling from the hybrid generative model defined by Figure 2 and the interaction model $p_\theta$. A notable difference between the augmented training set $\overset{+}{\mathcal{D}}$ and the original training set $\mathcal{D}$ is that the former contains ground truth values for the expert's variables $z_e$. As we assume that the interaction model is $\overset{+}{\Omega}$-*exact*, we freeze it and only fine-tune the encoder $q_\psi$ on $\mathcal{D} \cup \overset{+}{\mathcal{D}}$. We use a combination of the loss function $\ell$ of the original algorithm (e.g., Equation (4) for HVAE, and the Lagrangian of Equation (2) for APHYNITY) and a supervision on the latent variable objective to learn a decoder

---

**Algorithm 1** Expert augmented hybrid learning

1: $\mathcal{D} := \{(x^{(i)}, y^{(i)})\}_{i=1}^N \in (\mathcal{X} \times \mathcal{Y})^N$ ▷ A training set
2: $q_\psi(z_a, z_e|x, y)$ ▷ A parametric encoder
3: $p(y_e|x, z_e)$ ▷ An expert model
4: $p_\theta(y|x, y_e, z_a)$ ▷ A parametric interaction model
5: $l(x, y, \theta, \psi)$ ▷ A hybrid learning objective function
6: $p_+(z_e)$ ▷ A prior distribution on $z_e$ that covers both train and test scenarios
7: **procedure** TRAINING
8: $\quad \psi^\star, \theta^\star \leftarrow \arg\min_{\psi,\theta} \mathbb{E}_{(x,y)\sim\mathcal{D}} [l(x, y, \theta, \psi)]$
9: $\quad \theta^\star$ is frozen.
10: $\quad \overset{+}{\mathcal{D}} \leftarrow \text{GENERATEAUGMENTEDSET}$
11: $\quad \mathcal{D} \leftarrow \overset{+}{\mathcal{D}} \cup \mathcal{D}$
12: $\quad$ Finetuning the encoder on the augmented training set:
13: $\quad \psi^\star \leftarrow \arg\min_\psi \mathbb{E}_{\mathcal{D}} [l(x, y, \theta^\star, \psi)]$
$$\quad\quad - \mathbb{E}_{\overset{+}{\mathcal{D}}} [\log q_\psi(z_e|x, y)]$$
14: $\quad$ **return** $\psi^\star, \theta^\star$
15: **end procedure**
16: **procedure** GENERATEAUGMENTEDSET
17: $\quad \overset{+}{\mathcal{D}} \leftarrow \{\}$
18: $\quad$ **for each** $(x_o, y_o) \in \mathcal{D}$ **do**
19: $\quad\quad z_a \sim q_{\psi^\star}(z_a, z_e|x_o, y_o)$
20: $\quad\quad z_e \sim p_+(z_e)$
21: $\quad\quad y_e \sim p(y_e|x, z_e)$
22: $\quad\quad y \sim p_{\theta^\star}(y|x, y_e, z_a)$
23: $\quad\quad \overset{+}{\mathcal{D}} \leftarrow \overset{+}{\mathcal{D}} \cup \{(x_o, y, z_e)\}$
24: $\quad$ **end for**
25: $\quad$ **return** $\overset{+}{\mathcal{D}}$
26: **end procedure**

that solves

$$\psi^\star = \arg\min_\psi \mathbb{E}^+_{p(z_e)p(z_a)p(x)p(y|x,z_e,z_a)} \left[\ell(x,y;\theta,\psi) - \log q_\psi(z_e|x,y)\right],$$

$$(5)$$

$$\approx \arg\min_\psi \frac{1}{N + \overset{+}{N}} \sum_{(x,y)\in\mathcal{D}\cup\overset{+}{\mathcal{D}}} \ell(x,y;\theta,\psi) - \frac{1}{\overset{+}{N}} \sum_{(z_e,x,y)\in\overset{+}{\mathcal{D}}} \log q_\psi(z_e|x,y).$$

In our experiments we use a Gaussian distribution for the posterior, which is equivalent to a mean squared error (MSE) loss on the physical parameters. We provide a detailed description of the expert augmentation scheme in Appendix A. In principle, the regularizers (the norm of $f_a$ for APHYNITY and $R_{PPC}$, $R_{DA,1}$, and $R_{DA,2}$ for the Hybrid-VAE) are not necessary as Equation (5) only aims to improve the encoder whereas these terms aim to regularize the interaction model. However, in practice, we have observed that this does not matter and makes the implementation even more straightforward.

As a side note, we would like to emphasize the difference between the data augmentation proposed in this paper and the one from Takeishi & Kalousis (2021). While HVAE also requires to sample new physical parameters $z_e$, it is only to ensure that a sub-part of the encoder is able to infer correctly $z_e$ given $y_e$. This augmentation does not contribute robustness to distribution shifts on $y$ in contrast to ours.

## 4    Experiments

We assess the benefits of expert augmentation on three synthetic problems and one real-world experiment that are described by the ODE

$$\frac{dy_t}{dt} = F_e(y_t; z_e) + F_a(y_t; z_a),\qquad(6)$$

where $F_e : \mathcal{Y}_t \times \mathcal{Z}_e \to \mathcal{Y}_t$ is the expert model and $F_a : \mathcal{Y}_t \times \mathcal{Z}_a \to \mathcal{Y}_t$ complements it. In our notation $X$ is the initial state $y_0$ and the response $Y$ is the sequence of states $y_{1:t_1} := [y_{i\Delta t}]_{i=1}^{t_1/\Delta t}$. For all experiments we train the models to maximize $p_{\theta,\psi}(y = y_{1:t_1}|x = y_0)$ on the training data. We validate and test the models on the predictive distribution $p(y = y_{1:t_2}|x = y_0, x_o = y_0, y_o = y_{1:t_1})$, where $t_2 > t_1$ assesses the generalization over time. A brief description of the different problems is provided below.

### 4.1    Synthetic experiments

**The damped pendulum** is often used as an example in the hybrid modelling literature (Yin et al., 2021; Takeishi & Kalousis, 2021). The system's state at time $t$ is $y_t = \begin{bmatrix} \theta_t & \dot\theta_t \end{bmatrix}^T$, where $\theta_t$ is the angle of the pendulum at time t and $\dot\theta_t$ its angular speed. The evolution of the state over time is described by Equation (6), where $z_e := \omega$, $z_a = \alpha$ and

$$F_e := \begin{bmatrix} \dot\theta_t & -\omega_0^2 \sin\theta_t \end{bmatrix}^T \quad \text{and} \quad F_a := \begin{bmatrix} 0 & -\alpha\dot\theta_t \end{bmatrix}^T.\qquad(7)$$

The corresponding systems are defined by the damping factor $\alpha$ and $\omega_0$, the fundamental frequency of the pendulum.

**The RLC series circuits** are electrical circuits made of 3 electrical components that may model a large range of transfer functions. A schematic of such circuit is shown in Figure 10. These models are often used in biology (e.g., the Hodgkin-Huxley class of models (Hodgkin & Huxley, 1952), in photoplethysmography (Crabtree & Smith, 2003)) and in electrical engineering to model the dynamics of various systems. The system's state at time $t$ is $y_t = \begin{bmatrix} U_t & I_t \end{bmatrix}^T$, where $U_t$ is the voltage over the capacitor and $I_t$ the current in the circuit. The evolution of the state over time is described by Equation (6), where $z_e := \{L, C\}$, $z_a = \{R\}$ and

$$F_e := \begin{bmatrix} \frac{I_t}{C} \\ \frac{1}{L}(V_t - U_t) \end{bmatrix} \quad \text{and} \quad F_a := \begin{bmatrix} 0 \\ -\frac{R}{C}I_t \end{bmatrix}.\qquad(8)$$

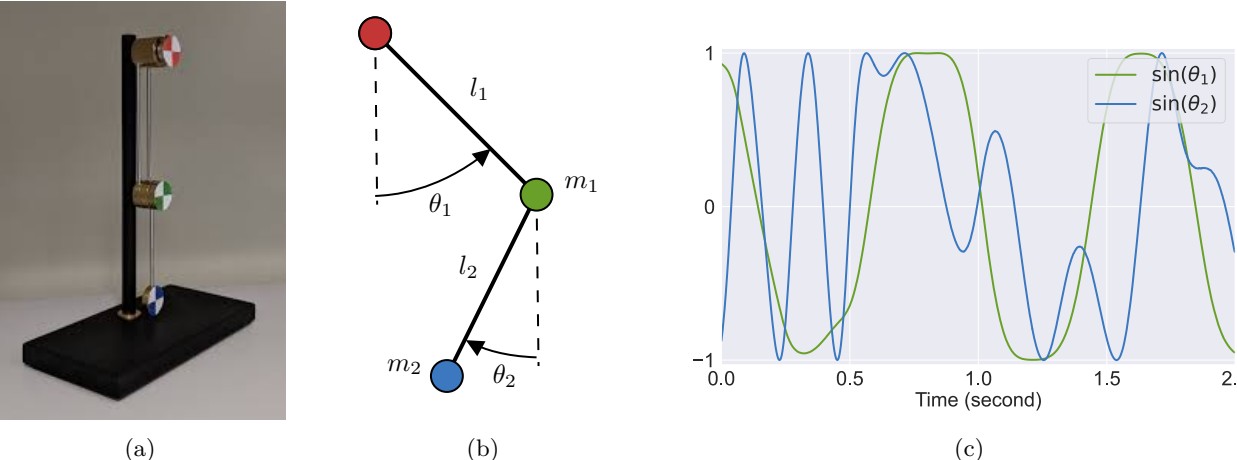

Figure 4: The double pendulum setup. (a) A photograph of the double pendulum at rest, reproduced from Asseman et al. (2018). (b) A simplified sketch of the setup. (c) An example of the time series extracted from the videos of the double pendulum.

The dynamics described by the RLC circuit is more diverse than for the pendulum and the system can be hard to identify. This system is characterised by the resistance $R$, capacitance $C$, and inductance $L$, provided $V_t$ is known.

**The 2D reaction diffusion** was used by Yin et al. (2021) to assess the quality of APHYNITY. It is a 2D FitzHugh-Nagumo on a $32 \times 32$ grid. The system's state at time $t$ is a $2 \times 32 \times 32$ array $y_t = \begin{bmatrix} u_t & v_t \end{bmatrix}^T$. The evolution of the state over time is described by Equation (6), where $z_e := \{a, b\}$, $z_a = \{k\}$ and

$$F_e := \begin{bmatrix} a\Delta u_t \\ b\Delta v_t \end{bmatrix} \quad \text{and} \quad F_a := \begin{bmatrix} R_u(u_t, v_t; k) \\ R_v(u_t, v_t) \end{bmatrix}, \tag{9}$$

where $\Delta$ is the Laplace operator, the local reaction terms are $R_u(u, v; k) = u - u^3 - k - v$ and $R_v(u, v) = u - v$. This model is interesting to study as it considers a state space for which neural architectures may have a real advantage compared to other ML models. In particular, convolutional neural networks are effective for processing signals with spacial and/or temporal correlation.

In these experiments we analyze the effect of our data augmentation strategy on APHYNITY and HVAE. All models explicitly use the assumption that the interaction model follows the structure of Equation (6). For each problem the validation and test sets are respectively IID and OOD with respect to the training distribution. The best models are always selected based on validation performance, that is with samples from $\Omega$. We provide additional details on the different expert models, dataset creation, and neural networks architectures in Appendix C.

## 4.2 A real world dataset - the double pendulum

We next validate the benefit of the expert augmentation in a controlled real-world setting. The dataset of a double pendulum introduced by Asseman et al. (2018) contains 21 videos of the pendulum shown in Figure 4a. Each run lasts approximately 40 seconds and is recorded at 400Hz. We can extract the position of the pendulum limbs from each frame with elementary computer-vision tools. Each recording starts from different initial conditions, leading to many states of this chaotic system. For illustration, we showcase the evolution of the arms' angles over time in Figure 4c.

We sketch a simplified representation of the double pendulum in Figure 4b. Its state is a four-dimensional vector $y_t = \begin{bmatrix} \theta_1(t) & \theta_2(t) & \dot{\theta}_1(t) & \dot{\theta}_2(t) \end{bmatrix}^T$, containing the position and speed of both masses. We can derive

(e.g., (Stachowiak & Okada, 2006)) the kinetics of the frictionless pendulum from first-principle physics,

$$\ddot{\theta}_1 = \frac{-g(2m_1 + m_2)\sin\theta_1 - m_2 g\sin(\theta_1 - 2\theta_2) - 2\sin(\theta_1 - \theta_2)m_2(\dot{\theta}_2^{\,2}l_2 + \dot{\theta}_1^{\,2}l_1\cos(\theta_1 - \theta_2))}{l_1(2m_1 + m_2 - m_2\cos(2\theta_1 - 2\theta_2))}, \tag{10}$$

$$\ddot{\theta}_2 = \frac{2\sin(\theta_1 - \theta_2)(\dot{\theta}_1^{\,2}l_1(m_1 + m_2) + g(m_1 + m_2)\cos\theta_1 + \dot{\theta}_2^{\,2}l_2 m_2\cos(\theta_1 - \theta_2))}{l_2(2m_1 + m_2 - m_2\cos(2\theta_1 - 2\theta_2))}. \tag{11}$$

This ODE is a suitable expert model candidate for a real-world double pendulum.

We assume that $m_1 = m_2$. Therefore the effect of masses reduces to constant values in the expert ODE. The length of the two arms are known, $l_1 = 91mm$ and $l_2 = 70mm$. The total energy of the double pendulum decreases over time in all videos, which lets us speculate about frictions, not explained by the expert model. In addition, the expert model does not consider potential vibrations or errors in extracting the arms' positions. Hybrid learning has the potential to correct these mispecifications automatically. In comparison, the characterisation of the frictions from first-principle physics is challenging and is still a research subject (Aghili, 2020).

Similarly to the damped pendulum, we consider the initial angular positions, $\theta_1(t = 0)$ and $\theta_2(t = 0)$, known. The encoder must predict the initial angular speeds $z_e := \{\dot{\theta}_1(t = 0), \dot{\theta}_2(t = 0)\}$ which are the only free parameters of the expert model. The encoder only observes $\theta_1$ between $t = 0ms$ to $t = 100ms$ and $\theta_2$ between $t = 50ms$ to $t = 100ms$ which makes the estimation of $z_e$ complicated. Then, we predict the angular positions between $t = 0$ and $t = 200ms$ given $\theta_1(t = 0)$ and $\theta_2(t = 0)$ and the estimation of $z_e := \{\dot{\theta}_1(t = 0), \dot{\theta}_2(t = 0)\}$ via the hybrid decoder.

We create a dataset with many initial conditions by splitting the videos into consecutive chunks of 20 frames sub-sampled at 100Hz, i.e., 200ms of video. We construct a distribution shift, as shown in Figure 11 from Appendix C.5, over the expert variables $z_e$ by splitting each 40 seconds sequence into three parts. The training set only contains chunks from the last 16 seconds of each run. It corresponds to configurations with smaller energy and, thus, slower angular speeds than the test set, which only contains frames from the first 12 seconds. The validation set contains the remaining 12 seconds of frames in the middle.

### 4.3 Results

**Performance gain from augmentation.** *This experiment demonstrates that HVAE and APHYNITY are not robust to OOD test scenarios in opposition to the corresponding AHMs, as shown in Figure 1* for the 2D diffusion problem and in Appendix D for the two other problems. We emphasize that our intention is not to declare a winner between HVAE and APHYNITY. Indeed, both algorithms have already demonstrated performance superior to black box ML models. Hence, we only report a very simple baseline that is the mean value of the signals. We want to compare performance in OOD settings and empirically validate the benefit of AHMs. We compare the predictive performance in Figure 5 (see Table 1 for the raw numbers). Although classical hybrid learning strategies do very well on the IID validation set, they exhibit poor generalization on OOD test sets for all three problems. We also observe some disparity between APHYNITY and HVAE. In addition to different learning strategies, this is probably due to differences in the networks' architectures as they were respectively inspired from the corresponding pendulum experiment in each paper. However, even if one method may outperform the other for some problems, they both benefit from our augmentation strategy (APHYNITY+, HVAE+). Overall, the effect of augmentation goes up to dividing the test error by a factor of $e^{4.6} \approx 100$ in some cases.

**Stability for non-exact models.** The empirical results from Figure 5 are very important as they show that even when the decoder is not $\Omega$-*exact* (and hence not $\overset{+}{\Omega}$-*exact*), augmentation may still work. In particular, Figure 6 shows that the encoder does not predict the physical parameters perfectly. This indicates that the encoder is not $\Omega$-*exact* and neither should be the decoder. This plot shows the relative error on the physical parameters computed as $\sum_{i=1}^{k} \frac{1}{k} \left| \frac{z_e^i - \mu_\theta^i}{z_e^i} \right|$, where $\mu_\theta^i$ is the estimated most likely value of the $i^{th}$ component of the physical parameters. We first notice that APHYNITY and HVAE perform differently and their performance depends on the specific problem. While APHYNITY accurately estimates the physical

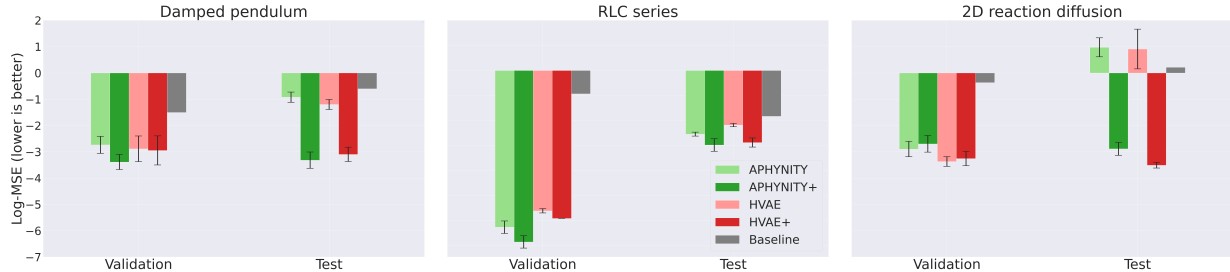

Figure 5: The average log-MSEs over 10 runs for three synthetic problems on the validation and test sets. We compare HVAE (in red) and APHYNITY (in green), in light colours, to their expert augmented versions HVAE+ and APHYNITY+, in darker colours. *On the test sets, AHMs outperform the original models, and by a large margin on the pendulum and diffusion problems. Moreover, augmentation conserves the relatively good performance on the validation set (IID w.r.t. the training set).*

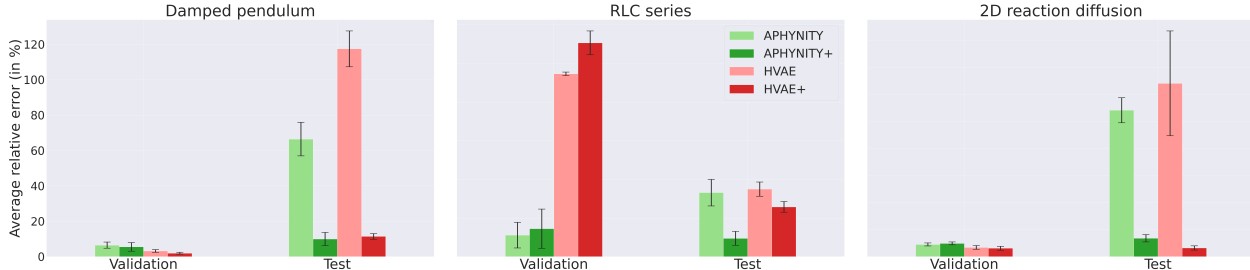

Figure 6: Comparison of mean relative precision (in %, ± indicates one standard deviation) over 10 runs of predicted physical parameters of different hybrid modelling strategies in validation and OOD test settings. Augmented versions are denoted with a +. *While the accuracy of APHYNITY and HVAE is good on the validation set, it collapses on the OOD test set. On the opposite, the augmented versions perform well on both validation and test sets.*

parameters on the IID validation set for the 3 problems, HVAE's performance are mixed on the RLC problem as it makes prediction that are around 120% away from the nominal parameter value on average whereas APHYNITY reduces this error to 6%. Interestingly, we observe that the proposed augmentation strategies improve the encoder such that it accurately estimates the physical parameters also on the OOD test set even for HVAE on the RLC problem. This confirms that the augmentation strategy is helpful even when the hybrid model is not $\Omega$-*exact*. As a conclusion, augmented hybrid learning outperforms classical hybrid learning both on the predictive accuracy and at inferring the expert variables.

**Effect of out of expertise shift.** *This experiment supports that our augmentation strategy may remain beneficial even when the train and test supports of $z_a$ are not identical.* This scenario corresponds to samples $(x, y)$ generated by $(z_a, z_e) \in (\overset{*}{\tilde{\mathcal{Z}}}_a \setminus \mathcal{Z}_a) \times \tilde{\mathcal{Z}}_e$ depicted by the violet domains in Figure 3. In Figure 7 we observe the log-MSE of augmented and non-augmented hybrid models trained for $(z_a, z_e) \in \mathcal{Z}_a \times \mathcal{Z}_e$ on test data that are generated with $(z_a, z_e) \in \tilde{\mathcal{Z}}_a \times \tilde{\mathcal{Z}}_e$. For the pendulum, the support over $z_a = \alpha$ is $[0, 0.3]$ in train and $[0.3, 0.6]$ in test; For the 2D reaction diffusion, $z_a = k$ is $[0.003, 0.005]$ in train and $[0.005, 0.008]$ in test. We observe that augmented models outperform the original models by a large margin. These results suggest that augmentation is valuable even when the distribution shift is not caused by the expert variables. However, if the shift on $z_a$ becomes the dominant effect, augmented models also eventually becomes vulnerable to shifts on $z_e$ as demonstrated by supplementary experiments in Appendix C.

**Real-world double pendulum.** *This experiment demonstrates the potential effectiveness of expert augmentation for real-world applications* In Figure 9, we compare the empirical performance of APHYNITY(+) and HVAE(+) with two baselines: 1) *Expert only*: the ODE of a friction-less double pendulum, 2) *ML only*: an agnostic neural ODE. We observe in Figure 9a that the augmentation improves the validation and test predictive performance by a non-negligible margin, as confirmed visually by Figure 8. In addition, the augmentation improves the estimation of the expert parameters by up to a factor of two for APHYNITY in

the OOD test scenarios, as shown in Figure 9b. In order to achieve these results, we finetune the encoder on expert augmentations with $\dot{\theta}_1 \sim \mathcal{U}[-15, 15]$ and $\dot{\theta}_2 \sim \mathcal{U}[-30, 30]$.

We have observed that optimizing the HVAE on the double pendulum data eventually becomes numerically unstable along training. It is why we did not manage to obtain HVAE's performance on par with APHYNITY and the ML only baseline. In consequence, the expert augmentation is also unstable if it relies on the best model. We circumvent these numerical issues by applying the expert augmentation to an earlier version of the HVAE model. The expert augmented model HVAE+ outperforms the best HVAE model on predicting the trajectory as seen in Figure 9a. Finally, we study the effect of applying the expert augmentation at various stages during training for APHYNITY in Appendix D.4. We observe that expert augmentation may reduce the gap between the train/validation/test performance on the double pendulum even when the interaction model is not fully trained. This result hints again that expert augmentation can help even when the interaction model is not learned perfectly.

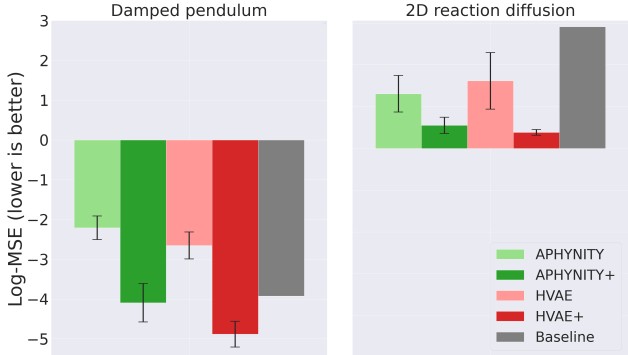

Figure 7: The average log-MSEs over 10 runs for the *damped pendulum* and *2D reaction diffusion* problems on a test distribution for which $z_a$, in addition to $z_e$, is also shifted. *AHM achieves better peformance than standard algorithms even when the test distribution support $z_a$ differs from the training.*

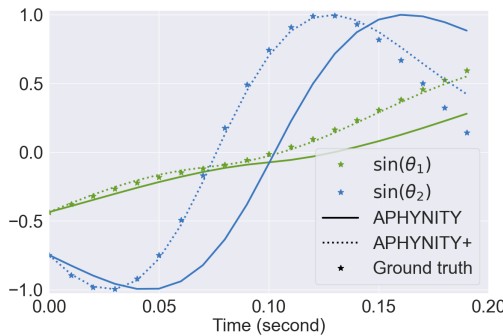

Figure 8: A cherry-picked example of the predicted angular positions of the double pendulum. *We observe that the proposed expert augmentation allows the hybrid model to predict more accurately the state of the double pendulum in the future than the non-augmented hybrid model.*

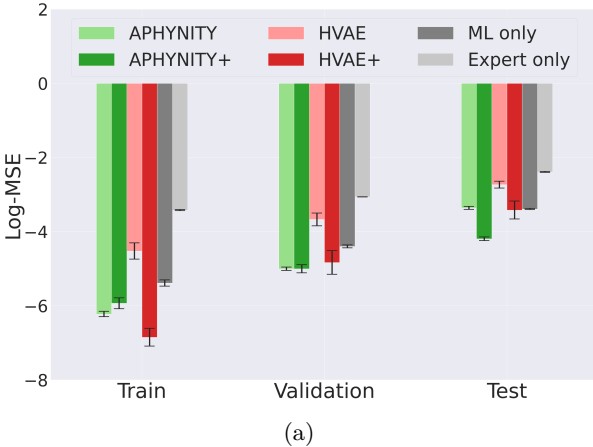

(a)

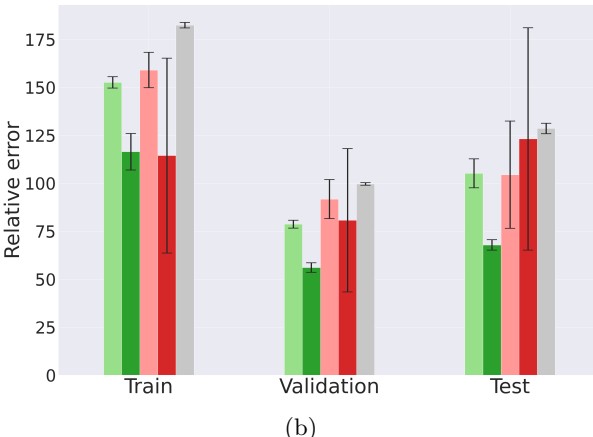

(b)

Figure 9: The results of the double pendulum experiment. (a) The average log-MSEs over three experiments. The baselines rely either only on the expert ODE or a neural ODE to predict the pendulum's state. *The proposed expert augmentation slightly reduces the predictive performance on the training set for APHYNITY but increases the generalisation capabilities of both APHYNITY and the HVAE. APHYNITY+ outperforms the baselines on all sets.* (b) The average relative errors on the initial angular speeds over three runs. *The proposed expert augmentation improves the accuracy of the physical parameters estimation both in the IID and OOD settings for APHYNITY.*

# 5 Related work

## 5.1 Hybrid modelling

Hybrid Learning, or gray box modelling as called in its early days in the 90's (Psichogios & Ungar, 1992; Rico-Martinez et al., 1994; Thompson & Kramer, 1994; Rivera-Sampayo & Vélez-Reyes, 2001; Braun & Chaturvedi, 2002), has been a popular method to learn models that are both expressive and interpretable, while also allowing them to be learnt on fewer data. The interest for hybrid learning (Mehta et al., 2021; Lei & Mirams, 2021; Reichstein et al., 2019; Saha et al., 2021; Guen & Thome, 2020; Levine & Stuart, 2022; Espeholt et al., 2022) has greatly increased since the outbreak of recent neural network architectures that simplify the combination of physical equations within ML models. As an example, Neural ODEs (Chen et al., 2018) and convolutional neural networks (LeCun et al., 1995, CNN) are privileged architectures to work with dynamical systems described by ODEs or PDEs. While most of the literature focus on the predictive performance of hybrid models, recent work have also shown that this framework helps to infer the physical parameters accurately (Yin et al., 2021; Takeishi & Kalousis, 2021). This is aligned with Zyla et al. (2020) (see Section 40.2.2.2) which observe that inference on incomplete models results in a *systematic bias.* Similar to hybrid learning, they extend the model with *nuisance* parameters in order to improve its fidelity, and to reduce the systematic bias.

In this work, we decided to study Yin et al. (2021) and Takeishi & Kalousis (2021) for two reasons that distinguish them from the rest of the literature. First, these are notable examples of algorithms that can be applied to a broad class of problems in contrast to papers that focus on specific applications (Lei & Mirams, 2021; Reichstein et al., 2019). Second, those methods also learn a reliable inference model for the physical parameters, suggesting that the expert model is used properly in the generative model, which is a key assumption for our augmentation. While Takeishi & Kalousis (2021) claim to achieve robustness, we argue that this statement is incomplete as HVAE fails in OOD settings. In particular, their approach is only able to generalize with respect to unseen time or initial state if the model correctly identifies the latent variables $z_a, z_e$. HVAE cannot generalize to new physical parameters because the encoder's validity is bound to the training set for the physical parameters.

## 5.2 Combining hybrid modelling and data augmentation

Close to our idea is the one proposed in Shrivastava et al. (2017) where they train a GAN model that improves the realism of a simulated image while conserving its semantic content (e.g., eyes colour) as modeled by the simulation parameters. The generated data with their annotations may then be used for a downstream task, such as inferring the properties of real images that corresponds to simulation parameters. The GAN objective from Shrivastava et al. (2017) requires that the two distributions induced by the semantic content of real and simulated data are identical. On the opposite, we consider training data that corresponds to expert parameters with limited diversity, and overcome this scarcity with expert augmentation. Another line of work similar to ours is Sim2Real, which considers the task of transferring a model trained on simulated data to real world (Doersch & Zisserman, 2019; Sadeghi et al., 2018a;b). Robust hybrid learning, as a way to enhance simulations, could be used for Sim2Real.

## 5.3 Robust ML and invariant learning

Various statistical methods have been introduced to ensure models generalize under distribution shift. Domain-adversarial objectives aimed at learning (conditionally) invariant predictors (Ganin et al., 2016; Zhang et al., 2017; Li et al., 2018), GroupDRO (Sagawa et al., 2019) optimizing for worst-case loss over multiple domains and IRM (Arjovsky et al., 2019) as well as sub-group calibration (Wald et al., 2021) aiming to satisfy calibration or sufficiency constraints to learn features invariant across domains. Extensions, able to infer domain labels from training data have been proposed as well (Lahoti et al., 2020; Creager et al., 2021), partially inspired by fairness objectives (Hébert-Johnson et al., 2018; Kim et al., 2019). In contrast to AHM, all of these methods rely on the variation of interest being present in the training data.

## 6    Discussion

We now discuss the potential limitations of our method and its underlying assumptions.

**Erroneous interaction model.**    The exactness of the hybrid component $p_\theta(y|x, y_e, z_a)$ is a critical assumption underlying our expert-based augmentation strategy. Unfortunately, this component is learned from training data only. Hence, we cannot prove its exactness on the test domain in the general case as the input domain over $\mathcal{Y}_e$ may be different from the training. However, we argue that assumptions on the class of interaction model may alleviate this problem. As an example, we might consider to chose the best interaction model over a fixed set of potential interaction. If the correct interaction is present in this set we should eventually select it from data. A less extreme example is when we consider an additive hybrid model and embed this hypothesis into the interaction model, generalization to unseen $y_e$ follows as long as the range of values is the same as in the training set. If this assumption is too strong, we could still expect that $p_\theta(y|x, y_e, z_a)$ generalizes to unseen $y_e$ because hybrid learning drives $y$ samples from $p_\theta$ to be close to $y_e$. It implies that the corresponding function approximator is stable, which helps generalizing to unseen scenarios. However, we must acknowledge that we cannot guarantee the stability of the interaction model in the general case. In practice, finding a good inductive bias for the interaction model is important. Moreover, it is usually easier to embed an effective inductive bias in the interaction model rather than in the encoder. Indeed, the parameter identification (the encoder $q_\psi$) is often less-well understood or more complex than the generative model itself.

**Diagnostic.**    While crucial, we cannot guarantee the exactness of the decoder $p_\theta$ in general because we only evaluate the encoder and the decoder jointly on data points $(x, y, x_o, y_o)$. However, in some cases we can detect model misspecification by observing that the predictive model $p_{\theta,\psi}(y|x, x_o, y_o)$ is imperfect. Making this observation is not always simple as it requires prior knowledge on the expected accuracy of an exact model. However, when the system is identifiable, we may argue that the accuracy should be only limited by the intrinsic measurement noise on $y$.

**Relaxing exactness.**    Even with a solid inductive bias on the decoder, achieving exactness is hard in practical settings. However, our experiments demonstrate that expert-augmentation works in practice. We can explain this by looking at Figure 3. If the generative model that maps $x$ and $(z_a, z_e)$ is incorrect, the mapping from $\mathcal{Z}_a$ and $\mathcal{Z}_e$ could be slightly off from $\overset{+}{\Omega}$. However, this does not preclude the set of augmented samples from being closer to $\overset{+}{\Omega}$ than $\Omega$ and from inducing a better predictive model on $\overset{+}{\Omega}$ than the original model trained only on $\Omega$. Another argument is the effectiveness of data augmentation for training classical deep learning models, which works well even when some augmentations do not generate realistic samples. In addition, Zhang et al. (2022) have observed that the dominant cause of overfitting in classical variational autoencoders is usually coming from the encoder, similarly to what happens with amortized hybrid learning. In this regard, an unexpected advantage of expert augmentation would be to reduce this source of overfitting in addition to the OOD robustness gain.

**Limitations.**    We have considered expert models that are parameterized by a small number of parameters and are covered densely via sampling. For higher dimensional parameter space the augmentation strategy might become inapplicable. Hence, a more ingenious sampling strategy, such as worst-case sampling, would be required. Another difficulty is choosing a plausible range of parameters that contains both the train and the test support; this will often need a human expert in the loop. If the chosen distribution does not cover some test configurations the robustness guarantees for these configurations collapse. On the opposite, a distribution that is too broad may also impact negatively the quality of the model. For instance, learning the right encoding behaviour for those unnecessary configurations consumes representation capacity that is then unavailable for IID configurations. In this case the model might perform worse than before augmentation on the training distribution. Indeed, some of our results have shown that the training error may slightly increase while the test error decreases.

In addition, we assume that the train distribution of $z_a$ is representative of the test distribution. We empirically observed that a softer version of this assumption could be enough. However, performance will

eventually decline as the support of the test distribution for $z_a$ is far from the training domain. Finally, we have only validated our expert augmentation for amortized inference settings. Nevertheless, online inference algorithms, such as Markov-chain Monte Carlo, also rely on hyperparameters (Campbell et al., 2021) or learnt distributions (Brofos et al., 2022) which must be tuned to the specific problem of interest. They could also benefit from expert augmentation if those actionable components are contextually predicted by a ML model.

## 7 Conclusion

We have described hybrid learning within a probabilistic model in which one component of the latent process, denoted the expert model, is known. In this context, we have established that state-of-the-art algorithms are vulnerable to distribution shifts. Grounded in this formalisation, we have derived that expert augmentations induce robustness to OOD settings. We have shown on a set of synthetic settings and on a real-world system the favorable effect of expert augmentation on the OOD performance where standard hybrid learning algorithm fails. Finally, we have discussed how our assumptions transfer to real-world settings and have described potential shortcomings.

Our augmentation should benefit from future progress in hybrid learning as it shall apply to any amortized hybrid modelling algorithms. Providing more substantial constraints on the targeted hybrid model is an essential direction for further improving the robustness of hybrid models. For instance, the minimal description length principle (Grünwald, 2007) could be an excellent resource for investigating the balance between the model's capacity and robustness. In parallel, empirical and theoretical investigations of the learnt interaction model's properties in various practical settings would unlock new understandings of the capabilities and limitations of hybrid learning algorithms. In the context of expert augmentation, it should enable to do more realistic assumptions for its applicability and to derive more precisely the expected robustness gain. Future work is also needed to show how hybrid learning and expert augmentation translate into performance gain over classical ML on challenging real-world applications.

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
