# OpenReview forum: "Robust Hybrid Learning With Expert Augmentation"
_TMLR — Accepted by TMLR_

### Review · Reviewer_Wm5U · 2022-11-06

**Summary Of Contributions:**

This paper shows that hybrid learning algorithms (those that incorporate a learned parametric model and a domain-specific 'expert' model) can fail when applied to out-of-distribution data. The authors propose to leverage an assumption that expert models are still valid OOD, to provide "expert augmentation" which improved OOD performance.

**Audience:**

Yes

**Claims And Evidence:**

No

**Requested Changes:**

(In addition to [Critical] changes and those that would simply [Strengthen] the submission, I provide a few very [Minor] nitpick requests.)

## Evidence

A. [Critical] Extend the scope of real-world experimental evaluation. Having a single real-world evaluation is not enough to make claims about improved generalization capabilities. I would like to see additional experiments with real-world data. In particular, I would like to see results for cases in which the expert models are more complicated and have a larger number of parameters and cases in which the non-expert model is more complicated (i.e., I would like to see some slightly more modern and large NNs being used to solve complex tasks). It would be great to get some intuition for how the relative complexities of expert and non-expert components impact the performance of expert augmentation.

B. [Critical] Evaluate expert augmentations on HVAE for real-world problems. With only APHYNITY providing evidence in the real-world setting, it is unclear whether expert augmentations apply to general hybrid models in a real-world setting.

C. [Critical] Apply expert augmentations to additional algorithms. In my opinion, even applying expert augmentations to HVAE and APHYNITY is not enough. Alternatively/additionally, provide a more detailed discussion of other hybrid learning algorithms and how exactly they fall within the probabilistic framework presented in Figure 2 (i.e., how expert augmentations can be applied to these other algorithms). Adding as many experimental results as possible would help sell the generality of expert augmentations.

D. [Critical] Provide ablation studies and sensitivity analyses. In particular, I would like to see how the results/conclusions change/remain the same with different model architectures and hyper-parameters. I would also like to see the sensitivity to the sampling ranges/distributions of z_e (e.g., as described in Appendix B). Finally, what happens if the interaction model is not frozen during fine-tuning of the encoder?

E. [Critical] Provide further experimental details. In particular, further discussion about the choices of hyperparameters and the setup for hyperparameter optimisation (where applicable). For example, in appendix B.4, it is stated that an extensive hyperparameter search was required, but no details of that search are given.

F. [Strengthen] Provide a link to the code. In the appendix, there is a reference to more details being in the code, but not code was provided.

## Clarity

1. [Strengthen] I did not understand the sentence "Because the interaction between z_e and y is essentially defined by the expert model, it should be possible, and preferable, to learn an accurate predictive model of Y whose accuracy is independent from the training distribution of the expert variables z_e". Please provide some additional explanation/context.

2. [Strengthen] Provide a derivation for the Bayes optimal predictor in eqn 1.

3. [Strengthen] Please provide some discussion for considering only deterministic expert models. What is the motivation for this choice? What are the consequences? How does this affect generality? (e.g., this means that the KL divergence is not a good choice of distance metric.)

4. [Strengthen] It was slightly confusing that the differential equation for the ODE is written differently in eqns 2 and 3.

5. [Strengthen] Please provide some discussion for only considering distribution shifts in which the marginals for z_a and x are constant. How does this impact the generality of the work? How realistic/common is this setting in practice?

6. [Strengthen] The importance of assumption 1 was unclear to me. I am not sure what the formalism adds to the submission. Furthermore, it seems to me that assumption 1 is actually much stronger than the assumption that "p(y_e|x, z_e) is an accurate description of the system, thereby p_theta(y|y_e, x, z_a) should not be overly complex".

7. [Minor]  For the RLC circuit example. "U_t is the voltage around the capacitance" should instead be "U_t is the voltage over the capacitor". "V(t)" is never introduced. There is inconsistency in the notation for indicating time dependency: "U_t" and "I_t" vs. "V(t)". Perhaps a small circuit diagram would be helpful.

8. [Minor] Using "array" rather than "tensor" to describe multi-dimensional generalisations of scalars, vectors and matrices is preferred since "tensor" has its own meaning in Physics see https://stats.stackexchange.com/questions/198061/why-the-sudden-fascination-with-tensors.

9. [Strengthen] Why does the 2D reaction diffusion considering "a state space for which neural architectures may have a real advantage compared to other ML models"? What is the motivation for their potential advantage?

10. [Strengthen] Please motivate why "soft assumptions on the class of interaction model may alleviate this problem" (section 6).

**Strengths And Weaknesses:**

## Strengths

* This submission tackles an interesting and important problem. One might mistakenly assume that since hybrid systems incorporate expert knowledge, they are more robust than standard ML algorithms and do not suffer from reduced performance on OOD data. However, this is not the case, and work on addressing robustness in the specific context of hybrid systems is needed.

* This submission proposes a novel solution. Using a probabilistic lens, the authors propose to leverage the expert system further to improve robustness. As far as I can tell, both the author's perspective and proposal contain elements of novelty.

## Weaknesses

(I provide concrete requests to address the weaknesses in the following section.)

* The claims made in the submission are not well supported by evidence. While the submission does include some experimental evidence to support the claims, the experiments were not rigorous enough to give me confidence in the claims. The generality of the work is unclear to me. Furthermore, I have concerns about the reproducibility of the work.

* The submission is not written clearly enough. I struggled to understand some parts of the paper, leaving me with several unresolved questions.

---

### Review · Reviewer_yBPq · 2022-11-09

**Summary Of Contributions:**

This paper introduce a hybrid data augmentation strategy that can improve the generalization in the OOD setting.

**Audience:**

Yes

**Broader Impact Concerns:**

I didn't realize any broader impact on the ethical implications of the work.

**Claims And Evidence:**

Yes

**Requested Changes:**

1. Improve the presentation of the paper on the methodologies, see the weakness part.

1. In Figure 2, the X should be marked (like adding shadow) as an observed variable, otherwise the link between $Y_e\rightarrow Y$ will be unnecessary.

2. Why assume $p_\theta(y_e|x,z_e)$ is a delta distribution? It will make the density or the posterior density $p(z_e|x,y_e)$ ill-defined (not absolutely continous).

3. Using the sample pairs $(y_e,z_e)$ from the expert to train the encoder is very similar to the wake sleep training of the amortized inference[1]. Recent work also shows this can improve the in-disrtibution generalization of the encoder[2]. Maybe this connection can help the paper build a more principle explanation of the why the proposed method can improve the generalization in the OOD setting.


[1] Dayan P, Hinton G E, Neal R M, et al. The helmholtz machine. Neural computation, 1995
[2] Zhang M, Hayes P, Barber D. Generalization Gap in Amortized Inference. NeurIPS 2022

**Strengths And Weaknesses:**

# Strengths
The problem of combing expert knowledge and machine learning is interesting and the presented method shows improvements on several experiments.

# Weaknesses
The paper is not well-written and hard to follow, a lot of  necessary information is missing and the methodology introduction is unclear.
1. The main objective function in Section 3.1  ($l(x,y,\theta,\phi)$ + an encoder regularizer) relies on Equations 2 and 4.  Therefore, it is necessary to make a self-contained introduction of both equations and how they relate to the graphical model described in Figure 2. For example, the regularizers in equation 4 should be mathematically described.
2. In equation 4, there is a regularizer about ``marginal distribution of samples generated by the complete
model to be close to the marginal distribution that would be only generated by the physical model. '', it's not very clear why it makes sense in the OOD setting.
3. The added regularizer $-\log q_\phi(z_e|x,y)$ in section 3.1 is not very clear, the full objective should be presented so we can analyze the consistency.  For example, in the appendix the $z_e$ is sampled from the prior, so the objective should be written as an integration over the $p(z_e)$ etc..

---

### Review · Reviewer_EcRM · 2022-11-22

**Summary Of Contributions:**

In the context of hybrid modeling, where an expert mode is augmented by a spearate learned component, the authors propose a novel hybrid data augmentation strategy that improves performance on out-of-distribution scenarios (e.g., physical systems with initial conditions outside of the range encountered during training). The authors show that standard hybrid learning methods fail to learn robustly out-of-distribution. The authors evaluated their expert augmentation strategy on top of 2 different hybrid models (APHYNITY & HVAE), with they thoroughly evaluated it empirically on multiple synthetic domains (damped pendulum, RLC circuit, and 2D reaction diffusion) and data coming from a real-world double pendulum.

**Audience:**

Yes

**Broader Impact Concerns:**

No broader impact concern.

**Claims And Evidence:**

Yes

**Requested Changes:**

Suggestions:
 - Move the pseudo-code from Appendix A to the main text (in the form of a pseudo-code).
 - Rework the notions of $\Omega$-exactness (minor suggestion, it can be left as is)

**Strengths And Weaknesses:**

**Strengths**: This submission is very clearly written, and easy to follow even for an audience not familiar with the domain of hybrid learning. The experiments are thorough on multiple domains previously studied in the literature, as well as real-world data. The evaluation metrics are also very clear, and they show an overall boost in performance using the expert augmentation proposed here on both hybrid learning methods (APHYNITY & HVAE).

**Weaknesses**: While the background is very well presented, I think it's unfortunate that the authors don't expand on their contribution as much (Section 3.1), delegating the description of their expert augmentation (the core of their contribution) to the Appendix.

The method relies on the fact that one knows the true parameters $z_{e}$ of the expert augmented trajectories (to add a likelihood maximization term in the loss), which seems like a strong assumption. In particular, that's probably the reason why the authors had to acquire expert trajectories from a simulated (frictionless) environment for the augmentation on the real-world double pendulum experiment.

I found the use of $\Omega$-exactness a bit confusing. I understand that it unifies a lot of the concepts (exact over training and/or test settings), but it could probably be presented as one reads instead of having all concepts presented at once in Section 3. For example, in the "Stability for non-exact models" (Section 4.3), this could read "the decoder is not trained perfectly on the training data" instead of "the decoder is not $\Omega$-exact".

---

### Decision · Action_Editors · 2023-01-26

**Recommendation:** Accept with minor revision

**Comment:**

The reviewers were all on the fence for this paper.  They agreed that it provided scientifically interesting contributions and was novel.  There were initially concerns regarding clarity, but the authors have already improved these (and have promised to further improve them).  The reviewers did not seem to have any issues with the merits of the technical contribution.  However, there was a discussion about the empirical evidence presented in the paper.  Specifically, the reviewers felt that the experiments were still rather toy (e.g. double pendulum) and the baselines were too weak.  The authors have promised additional experiments i.e. ("We have also added an additional experiment that shows that expert augmentation robustly improves the shifted-test performance.", "We have added results on the double pendulum with HVAE.")

Ultimately, two of three reviewers felt the evidence and claims were justified but only one gave a (weak) accept and two voted for (weak) reject.  One reviewer found the evidence and claims were not justified, given the strength of the experiments.  Given the reviewer recommendations the paper is quite borderline for acceptance.

The majority of reviewers felt the evidence and claims were ok.  Stronger experiments would certainly strengthen the paper and ultimately make it more impactful.  However, the current (and promised) experiments should be sufficient to justify the claims made in the paper.  The reviewers seem to think the paper is correct, technically interesting to some audience of TMLR and innovative.  Following the guidance of TMLR (i.e. "Papers should be accepted if they meet the criteria, even if the contribution or significance of the work is modest.") this paper should be accepted.  I would ask the authors to continue to work on the clarity of the paper (i.e. they said "We have started to address some clarity issues raised by different reviewers" but they should finish doing so.) and incorporate the additional experiments.

**Audience:**

Hybrid modeling seems quite relevant to a sub-field in machine learning.  The authors claim that hybrid modeling can improve generalization and reduce misspecification, which are issues broadly affecting methods in ML.


**Claims And Evidence:**

The reviewers were split on claims and evidence.  Initially, multiple reviewers found that the authors over-claimed their contribution and felt that the experiments were too "toy" with too few baselines to justify the claims.  The author response convinced two reviewers that the claims and evidence were ok (partially due to the authors weakening their claims somewhat).  After discussion, one reviewer maintained that the experiments were too toy to justify publication.